# Modulatory Effects of NBF1, an Algal Fiber-Rich Bioformula, on Adiponectin and C-Reactive Protein Levels, and Its Therapeutic Prospects for Metabolic Syndrome and Type-2 Diabetes Patients

**DOI:** 10.3390/biomedicines10102572

**Published:** 2022-10-14

**Authors:** Nlandu Roger Ngatu, Mitsunori Ikeda, Daniel Kuezina Tonduangu, Severin Luzitu Nangana, Tomohiro Hirao

**Affiliations:** 1Department of Public Health, Kagawa University Faculty of Medicine, Kagawa 761-0793, Japan; 2Wellness and Longevity Center, University of Kochi, Kochi 781-8515, Japan; 3Department of Internal Medicine, Faculty of Medicine, University of Kinshasa, Kinshasa, Democratic Republic of the Congo; 4Preventive Medicine Unit, Kimbanguist Hospital of Kinshasa, Kinshasa, Democratic Republic of the Congo

**Keywords:** adiponectin, C-reactive protein, metabolic marker, sujiaonori

## Abstract

An unhealthy diet can lead to the development of metabolic disorders. C-reactive protein (CRP) has been reported to be an inflammatory component of metabolic disorders. Additionally, reduced adiponectin (APN) levels is known as a predictor of metabolic disorders. We report on the beneficial effects of NBF1, an algal fiber-rich formula, on CRP, APN, and diabetes markers. Additionally, associations between dietary nutrients, CRP, and APN were investigated. A dietary survey that used a brief self-administered diet history questionnaire, a validated 58-item fixed-portion-type questionnaire, and a 4-week placebo-controlled dietary intervention were carried out. The latter consisted of a twice daily intake of 3 g of sujiaonori alga-based powder as a supplement (NBF1, *n* = 16), whereas the placebo group received the same amount of corn starch powder (*n* = 15). CRP and APN levels were assayed by ELISA. Clinical cases comprising three subjects with metabolic disorders treated with NBF1, including two type 2 diabetes mellitus patients and one subject with hypercholesterolemia, are also reported. They received 2.1g NBF1 once daily for 12 weeks. Intakes of algal fiber and n-3 PUFA were positively associated with the increase of APN, and n-3PUFA intake was inversely associated with CRP. All cases had improved metabolic health profile.

## 1. Introduction

The global burden of lifestyle-related noncommunicable diseases (NCDs), such as metabolic and cardiovascular diseases, has been impacting health and productivity in almost all countries [1]. The nutritional patterns of individuals supports their daily physiological needs in nutrients for growth, development, and appropriate cell and tissue turnovers [2]. In addition, a healthy diet provides necessary energy and nutrients needed to sustain, build, and repair the body, and the food that supplies them may impact human health either positively or negatively depending on the quality and quantity of food intakes. It is well-established that poor nutrition or low-quality food consumption exposes people to the risk of chronic conditions, such as cardiovascular diseases (CVD), obesity, metabolic syndrome, diabetes, and cancer [3,4].

A study on the primary prevention of coronary heart disease showed that women who practiced a healthy diet and lifestyle pattern were 80% less likely to develop CVD [5]. Thus, assessing dietary nutrient intakes, as well as biological markers, may help to analyze physiological and pathological responses to diet and lifestyle patterns, identify subjects with nutritional deficiency and those at-risk of disease, and suggest dietary recommendations to restore optimal health status [2].

In addition to well-known cardiometabolic health markers, such as systolic blood pressure (SBP), diastolic blood pressure (DBP), circulating cholesterol levels, triglycerides (TG), fasting blood glucose (FBG), and body mass index (BMI) [6], recent investigations have demonstrated that circulatory C-reactive protein (CRP) [7,8,9] and adiponectin (APN) [10,11,12,13] are strong predictors of most prevalent cardiometabolic diseases, including metabolic syndrome (MetS), diabetes, hypertension, coronary heart disease, and atherosclerosis. Furthermore, recent studies have also suggested that CRP could be considered an independent marker of cardiovascular and metabolic disorders [14,15]. A study conducted in a sample of Japanese men and women during their hospital visits for health screening tests showed a stable and strong correlation between CRP level and metabolic syndrome (MetS), more importantly in women [16]. This finding suggests that CRP could be used as an inflammatory component of MetS. 

Japanese diet is reported to promote preserved insulin sensitivity, low diabetes prevalence, and healthy longevity [17]. Edible algal species are among food items commonly found in Japanese diet that exert beneficial health effects. In this report, we examined the association between the intake of diet and supplement (NBF1)-derived bioactive nutrients (fiber, polyunsaturated fatty acids) with APN and CRP. We also included case reports of T2DM patients and one subject with hypercholesterolemia who were supplemented with NBF1. 

## 2. Materials and Methods

### 2.1. Dietary Survey and Intervention with NBF1

#### 2.1.1. Participants and Sample Size of the Dietary Intervention

A dietary survey and a dietary intervention were conducted in Kochi prefecture, Japan, between 2016 and 2017. Participants were university students and staff who voluntarily accepted to take part in the study. The “Suji Aonori, Adiponectin, Cardiometabolic Risk and Skin Health Study” comprised a diet history survey and a randomized placebo-controlled intervention of 3 g NBF1, river alga (*Ulva prolifera*)-derived biomaterial, in apparently healthy subjects. Inclusion criteria were as follows: 20 years of age or older, and provision of a signed informed consent form. The full study protocol is available online at the WHO ICTRP trial registry [18].

Clinical examination of the participants was performed, and vital signs were monitored at baseline and at the end of the study. In addition, saliva sampling was carried out for the measurement of targeted bio markers associated with inflammation and metabolic health, namely CRP and APN. Of the 38 enrolled subjects, 32 subjects met the inclusion criteria. The only male subject who enrolled was excluded, and only data from 31 female participants (16 in NBF1 group, 15 controls) who completed the post-intervention dietary survey and underwent pre and post-intervention clinical examination and biospecimen sampling were considered in this report (Figure 1).

#### 2.1.2. Diet History Survey

The dietary questionnaire is a useful assessment tool of long-term dietary habit and food culture, which often vary by community and country. The brief self-administered diet history questionnaire (BDHQ) was used in this study. It is a validated 58-item fixed-portion-type questionnaire that has been developed at the University of Tokyo for assessment of Japanese diets [19,20]. The questionnaire uses the list of food and beverage items contained in the Japanese National Health and Nutrition Survey [21]; it comprises 102 questions on food items, portion sizes, and beverage items during the preceding month by considering the following five sections: frequency of food and nonalcoholic beverage items; daily rice and miso soup intake; frequency of drinking and the amount of alcoholic beverage intakes; cooking methods; general dietary behavior.

The crude values of energy and 42 selected nutrients are estimated based on the intake of food items, with the use of a purpose-built computer algorithm that uses standard tables of Food Consumption in Japan [22,23].

#### 2.1.3. Composition of the Supplements

The sujiaonori biomaterial-based supplement was made of 100% *Ulva prolifera* powder. This edible river alga is mass-cultivated in Aonori farms located in the Kochi and Tokushima prefectures in Shikoku, Japan. The samples used in this study were a gift from Kochi Ice Company, Japan. The term NBF1 combines the initial of the name of the author who initiated the “Bio Modulators of Adiponectin (BMA) Research project” in 2015, and initials from the term bio and formula. The supplement is rich in fiber (65% dry weight) made of ulvans [24]; it also contains approximately 14% of proteins, 11% of n-3 and n-6 polyunsaturated fatty acids (PUFA), and some flavonoids, carotenoids and minerals (1%). The control supplement was made of corn starch, to which was added Japanese spinach powder to give it a green-like color. Each study group received 3 g of the corresponding supplement twice a day before meals for four weeks.

### 2.2. Outcome Variables 

Body weight and height were measured to determine the body mass index (BMI) of each participant. Systolic (SBP) and diastolic blood pressure (DBP) were measured using an electronic BP monitor (Omron HEM-8712, Omron Co.Ltd, Kyoto, Japan). In addition to the above-mentioned clinical parameters, two bio makers associated with inflammation and cardiometabolic health were measured, namely CRP and APN.

### 2.3. Enzyme-Linked Immunosorbent Assay (ELISA) for Measurement of Bio Markers

CRP and APN were measured in saliva samples using the procedures described previously [25,26]. Briefly, at least 60 min prior to brushing teeth or drinking/eating in the morning, 5 mL of whole saliva was collected in prepared tubes with a funnel, at the biochemical laboratory of the Faculty of Nutrition, University of Kochi, Japan. After saliva sampling, the test tube was immediately put on ice and then stored at −20 °C in the laboratory. Bio markers were assayed using human adiponectin and C-reactive protein ELISA kits (AssyPro, St Charles, MO, USA) following the manufacturer’s instructions. These assays were performed in duplicate.

### 2.4. Clinical Case Reports

The case reports consist of three adult male subjects, including two T2DM patients and one subject at metabolic risk (hypercholesterolemia). The diabetic patients were cared for at diabetic clinics and received 2.1g NBF1 supplement once daily as an adjunctive treatment in combination with their usual oral antidiabetic drug. They were followed up and underwent blood sampling to measure cardiometabolic risk markers, namely fasting blood glucose (FBG), hemoglobin A1c (HbA1c), and lipids (triglyceride or TG, LDL-cholesterol, HDL-cholesterol), at baseline and three months later.

### 2.5. Data Collection and Analysis

Dietary survey data were managed by an independent research center that estimated values of energy and nutrients using a diet history software. Values of metabolic markers are expressed as means (SD). Clinical and biological data used in the statistical analysis were collected at the end of study. An unpaired *t*-test was used to compare mean values of clinical parameters and nutrient intakes, whereas a paired *t*-test was performed to compare the overall mean values of metabolic markers before and after dietary intervention among the three diabetic and metabolic risk cases. The normality of the distribution of the outcome variables (cardiometabolic markers) was assessed using the Shapiro–Wilk test. Linear regression analysis assessed the association between cardiometabolic markers (CRP, adiponectin, BMI, BP) and intakes of bioactive macro and micronutrients from the diet and supplements of both study groups in the dietary intervention study. Data analysis was performed with the use of Stata statistical software version 15 (Stata Corp., College Station, Texas, USA); *p*-values less than 0.05 were considered significant.

## 3. Results

### 3.1. Sociodemographic Characteristics, Dietary Intake of Macro and Micronutrients

Participants were females and nonsmokers. The mean age was 23.06 (2.21) years in NBF1 group and 23.06 (1.55) in the control group. In the NBF1 group, BMI was 21.11 (0.59) kg/m^2^ and 21.43 (0.60) in the control group. Daily energy intake was 1,371.9 ± 425.4 kcal in the NBF1 group and 1,356.12 ± 397.31 kcal/day in the control group. Regarding daily intake of macro nutrients, higher intakes of n-3 PUFA (3.24 (1.14) g/day, 95% CI: 2.99–4.61 vs. 2.34 (0.29), 95% CI: 1.21–2.98; *p* < 0.001) and fiber (12.08 (1.53) g/day, 95% CI: 11.93–13.44 vs. 9.63 (2.66), 95% CI: 8.45–10.01; *p* < 0.001) were observed in the NFB1 group compared with the control group (Table 1).

Additionally, when comparing both groups according to daily intake of other macro nutrients (protein of animal source, protein of plant source, n-6 PUFA) and micro nutrients (vitamin B6, vitamin B12, vitamin C, vitamin D, folates, α-tocopherol), the difference did not reach statistical significance. Regarding cardiometabolic markers, systolic blood pressure (SBP) was significantly lower in the NBF1 group (109.68 (2.04) mmHg, 95% CI: 103.92–112.25 vs. 110.53 (2.09), 95% CI: 105.04–114.02 in the control group; *p* < 0.05) (Table 1).

### 3.2. Relationship between Daily Algal Fiber and n3PUFA Intake and Cardiometabolic Markers

The linear regression analysis showed inverse associations between CRP and vitamin B6 (β = −12.8 (6.71), 95% CI: −22.91–201.81; *p* < 0.01), CRP and vitamin D (β = −149.5 (10.58), 95% CI: −267.31–32.69; *p* < 0.05), and also between CRP and NBF1-n-3 PUFA (β = −65.82 (2.81), 95% CI: −187.94–174.24; *p* < 0.05). Furthermore, APN was positively associated with vitamin D (β = 3.69 (2.95), 95% CI: 3.42–17.81; *p* = 0.034), NBF1-n-3 PUFA (β = 5.96 (5.06), 95% CI: 1.51–18.42; *p* = 0.038), and total NBF1-fiber (β = 10.51 (1.79), 95% CI: 5.79–18.23; *p* = 0.005). No association was observed between each of the two bio markers (CRP, APN) and vitamin B12, vitamin C, folates, total cholesterol, α-tocopherol, and protein levels (Table 2A).

Regarding the blood pressure (BP) profile of the study participants, DBP was inversely associated with NBF1-n-3 PUFA (β = −2.54 (0.39), 95% CI: −5.81–5.03; *p* = 0.042). However, no association was observed between SBP and any of the macro and micro nutrients that were considered in the data analysis; the same was true for BMI (Table 2B).

### 3.3. Effects of NBF1 Supplementation on Metabolic Markers in T2DM Patients and Subject with Metabolic Risk (Case Reports)

Figure 2A–E shows the plasma level values of fasting blood glucose (FBG), hemoglobin A1c (HbA1c), triglycerides (TG), low-density lipoprotein cholesterol (LDL-c), and high-density lipoprotein cholesterol (HDL-c) at baseline or day 1 of 2.1 g NBF1 supplementation (twice daily) and at 3 months later.
(1)Subject with hypercholesterolemia: 53 years old African healthcare worker with a history of high blood pressure, which has been successfully stabilized using antihypertensive drug (methyldopa). He had normal FBG and HbA1c, but high LDL-c level (140 mg/dL) at day 1 of NBF1 intake in 2020. An improvement of FBG (change: −15 mg/dL), HbA1c (change: −0.9%), TG −8 mg/dL), LDL-c (change: −23 mg/dL) were observed after a 3-month dietary intervention (Figure 2).(2)T2DM patient 1: 56 years old African living in Japan; he has been under oral thiazolidinedione for 2 years, before adding NFB1 in 2019. At baseline (day 1 of NBF1), fasting blood glucose (FBG) was 113 mg/dL, 8.3% for HbA1c, and elevated LDL-c (181 mg/dL) levels. Noticeable decrease of FBG (change: −18 mg/dL), HbA1c (−2.6%), TG (−6 mg/dL) and LDL-c (−12 mg/dL) were observed after 3 months of NBF1 supplementation (Figure 2).(3)T2DM patient 2: 57 years old African living in DR Congo; he has been under a combination of oral thiazolidinedione and insulin injection for 8 months, and NFB1 was added in 2019. Baseline FBG (153 mg/dL) and HbA1c (8%) were high. However, a marked improvement of metabolic health markers was observed, particularly for FBG (−49 mg/dL), HbA1c (−2.4%), TG (−13 mg/dL), and LDL-c (−7 mg/dL) after 3 months of NBF1 intake (Figure 2).

## 4. Discussion

Adipocyte-derived protein adiponectin plays an important role in human physiology and serves as a link among adiposity and cardiometabolic diseases, including T2DM and CVDs. The association between cardiometabolic disorders and hypo-adiponectinemia is well-documented [27,28,29,30], and many specialists suggest that the discovery of APN receptor agonists and/or APN modulators represents a big step towards preventing and, eventually, reversing the course of metabolic disorders associated with APN deficiency [31,32,33,34,35].

The present study showed that daily intake of fiber (over 80% of which came the from the NBF1 supplement) and n3 PUFA induced APN production more efficiently as compared with corn-starch-supplemented controls. Moreover, intakes of both fiber and n-3 PUFA from the marine algal biomaterial were positively associated with an increase in APN. Suji aonori grown in rivers and Aonori farms in western Japan is composed of 60–65% fiber, whose main compounds are ulvans; they are sulfated polysaccharides that possess a number of beneficial nutraceutical properties, such as antioxidative, immunostimulating, antiviral, and antihyperlipidemic activities [36,37]. Our experimental study also showed that ulvan from the river alga suji aonori has an anti-inflammatory activity that revealed to be more efficient than that of other algae found in the region [38].

The association between APN and n-3 PUFA has also been reported previously. Meta-analysis studies focusing on clinical trials related to the APN modulatory effect of PUFAs in diabetic patients have shown that n-3 PUFAs, such as EPA, DHA, or both in combination, increase APN production [39,40]. Our study also showed that n-3 PUFA was inversely associated with blood pressure, DBP in particular. A recent randomized clinical trial has also demonstrated that the daily intake of food rich in n-3 PUFA resulted in clinically relevant reductions in DBP [41]. Considering those reports and our findings, it is noteworthy that NBF1 might improve BP profile thanks to its APN modulating activity, possibly resulting from the combined effects of its ulvan-rich fiber and n-3 PUFA.

In recent decades, biomaterials from marine algae have been attracting the attention of many researchers. They are important ingredients in many of the natural products used in drug discovery and delivery, cosmetics, and health promotion. Sulfated polysaccharides, the major components of fiber from algae, have also been investigated for their anti-inflammatory and anticancer effects [39,42,43].

Clinical cases from our report also provide additional insights in terms of the health benefits of suji aonori biomaterial-based supplements in boosting cardiometabolic health. In fact, daily intake of NBF1 could improve a number of metabolic markers, namely FBG, HbA1c, and blood cholesterol profile, in subjects with T2DM and dyslipidemia. It is known that T2DM is associated with dyslipidemia [44]. Our observation is important, given that elevated LDL-cholesterol, as well as TG, are independent predictors of cardiometabolic diseases, and treatments that lower them have shown CVD benefits and reduction in related mortality.

This study did not explore the mechanisms underlying the APN modulating activity of the studied algal biomaterial. An experimental investigation that would elucidate those mechanisms has been conducted recently using animal model of obesity and T2DM; it has not been published yet. Additionally, only women took part in the main study; thus, there is a necessity to carry out similar investigations using male subjects. 

## 5. Conclusions

Notwithstanding the above-mentioned limitation, findings from this report provide a new understanding and scientific insights on the beneficial effects of NBF1, an algal biomaterial-based supplement, on the metabolic health of humans, as well as its potential to prevent and, possibly, reverse APN-associated cardiometabolic disorders.

## 6. Patents

Patent application was submitted in 2016 and a number has been recently issued online in Japan. IP FORCE (January 2022). “Sujiaonori-based health products”; No P2018-58770A. 

## Figures and Tables

**Figure 1 biomedicines-10-02572-f001:**
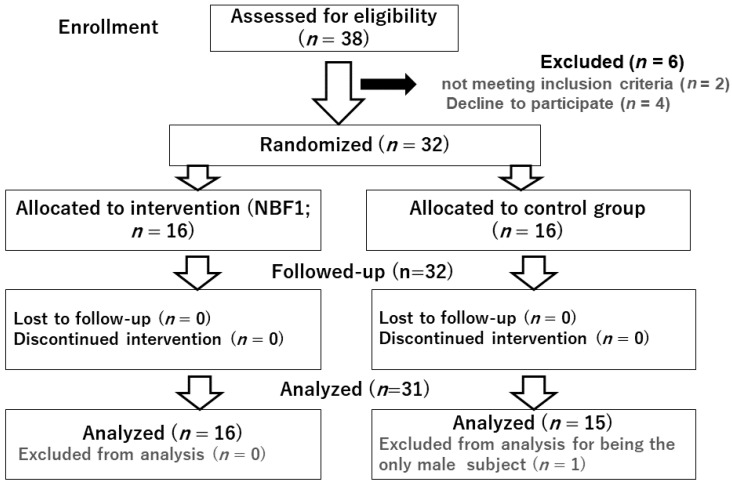
Diagram presenting the study participants sampling process.

**Figure 2 biomedicines-10-02572-f002:**
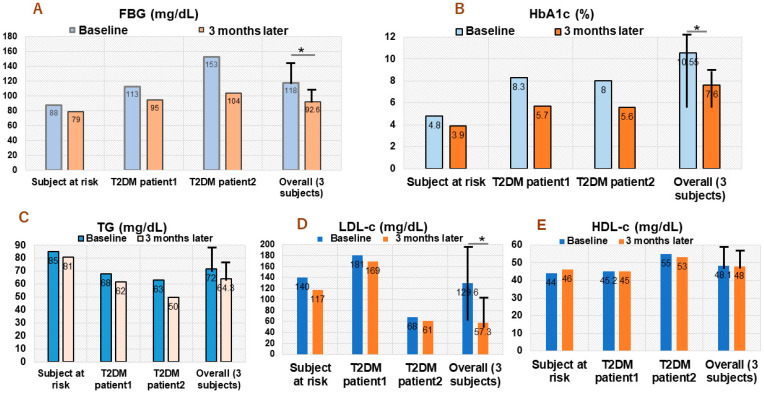
(**A**–**E**) Clinical cases: trend in changes of FBG, HbA1c, TG, LDL-c; HDL-c in two diabetic patients and one subject with hypercholesterolemia. FBG, fasting blood glucose; HbA1c, hemoglobin A1c; TG, triglycerides; LDL-c, low density lipoprotein cholesterol; HDL-c, high density lipoprotein cholesterol; T2DM, type 2 diabetes (*: *p* < 0.05).

**Table 1 biomedicines-10-02572-t001:** Clinical characteristics, total macro and micronutrients intake of the study participants.

Variables	Control (*n* = 16)		NBF1 Group (*n* = 15)		*p*-Value
	Mean (SD)	95% CI	Mean (SD)	95% CI	
Clinical parameters					
Age (years)	23.06 (1.55)	19.74–26.39	23.06 (2.21)	18.33–27.78	0.499
BMI (kg/m^2^)	21.43 (0.60)	20.14–22.73	21.11 (0.59)	19.84–22.3	0.353
Diastolic BP (mmHg)	73.26 (1.35)	70.35–76.17	73.5 (1.34)	70.63–76.36	0.451
Systolic BP (mmHg)	110.53 (2.09)	105.04 –114.02	109.68 (2.04)	103.92–112.25	0.046 *
Nutrients intake					
Vitamin B6 (mg/day)	1.07 (0.18)	0.69–1.46	0.781 (0.08)	0.59–0.97	0.08
Vitamin B12 (mg/day)	9.43 (2.6)	3.84–15.01	4.74 (0.46)	3.73–5.75	0.055
Vitamin C (mg/day)	86.74 (18.49)	57.08–136.40	85.31 (9.23)	87.17–135.44	0.166
Vitamin D (mg/day)	13.21 (3.94)	4.75–21.67	5.98 (0.59)	4.69–17.27	0.058
Total folates (mg/day)	224.44 (50.35)	176.43–392.44	215.66 (23.8)	163.77–267.56	0.125
α-tocopherol (mg/day)	6.02 (0.74)	4.42–7.61	6.22 (0.49)	3.94–6.11	0.145
n-3 PUFA (g/day)	2.34 (0.29)	1.21–2.98	3.24 (1.14)	2.99–4.61	0.000 *
n-6 PUFA (g/day)	7.81 (3.25)	6.02–9.61	8.75 (3.12)	5.35–10.98	0.218
Fiber (g/day)	9.63 (2.66)	8.45–10.01	12.08 (1.53)	11.93–13.44	0.000 *
Plant protein (g/day)	18.98 (3.15)	12.21–25.74	23.41 (1.77)	19.56–27.25	0.124
Animal protein (g/day)	29.918 (3.8)	21.75–38.08	23.45 (2.58)	17.87–29.03	0.088
Cholesterol (mg/day)	236.68 (19.14)	175.3–258.04	204.31 (38.5)	201.69–446.9	0.031 *

***** Statistically significant value: *p* < 0.05. BP, blood pressure; n-3 PUFA, omega-3 polyunsaturated fatty acid; n-6 PUFA, omega-6 polyunsaturated fatty acid; SD, standard deviation; NBF1, suji aonori biomaterial-based supplement.

**Table 2 biomedicines-10-02572-t002:** **A.** Associations between bio markers (CRP, APN) and bioactive nutrients.

**A**						
**Nutrients**	**CRP (pg/mL)**			**Adiponectin (ng/mL)**		
	**β** **(SE)**	**95% CI**	** *p* **	**β** **(SE)**	**95% CI**	** *p* **
Vitamin B6	−12.8 (6.71)	−22.91–201.81	0.023 *	0.14 (0.13)	−0.013–1.04	0.293
Vitamin B12	31.45 (10.27)	26.78–60.12	0.135	−7.88 (6.57)	−13.98–8.26	0.276
Vitamin C	−8.79 (2.27)	−81.74–64.15	0.755	1.43 (0.28)	0.26–2.11	0.182
Vitamin D	−149.5 (78.3)	−267.31–32.69	0.033 *	3.69 (2.95)	3.42–17.81	0.034 *
Folates	2.51 (10.58)	−22.69–31.71	0.688	0.65 (0.46)	−0.45–1.17	0.205
α−tocopherol	34.32 (20.17)	−159.1–228.72	0.283	26.4 (23.95)	−11.78–80.5	0.148
Cholesterol	0.05 (0.14)	−0.03–0.02	0.971	−1.55 (1.42)	−8.65–5.54	0.655
NBF1-n-3 PUFA	−65.82 (2.81)	−187.9–174.2	0.048 *	5.96 (5.06)	1.51–18.42	0.038 *
NBF1-n-6 PUFA	7.41 (4.29)	−2.54–17.42	0.112	0.68 (0.33)	−1.46–3.95	0.079
NBF1-fiber	−25.73 (43.97)	−122.52–71.06	0.057	10.51 (1.79)	5.79–18.23	0.005 *
Plant protein	−0.11 (0.01)	−0.003–0.05	0.313	0.13 (0.06)	−0.54–0.57	0.962
Animal protein	2.09 (0.12)	−0.25–3.09	0.808	−0.09 (2.22)	−0.57–1.37	0.678
**B**						
**Nutrients**	**DBP (mmHg)**		**SBP (mmHg)**		**Nutrients**	
	**β** **(SE)**	**95% CI**	** *p* **	**β** **(SE)**	**95% CI**	** *p* **
Vitamin B6	−2.48 (1.67)	−5.91–0.94	0.081	3.28(0.45)	−3.29–7.21	0.891
Vitamin B12	0.24 (0.20)	−0.18–0.67	0.258	0.25 (0.23)	−0.024–0.73	0.305
Vitamin C	−0.02 (0.08)	−0.06–0.01	0.278	−0.22 (0.03)	−0.09–0.51	0.838
Vitamin D	−0.02 (0.09)	−0.17–0.21	0.757	0.19 (0.15)	−0.13–0.54	0.243
Folates	−0.07 (0.06)	−0.02–0.05	0.232	−0.04 (0.02)	−0.07–0.37	0.471
α−tocopherol	−0.51 (0.39)	−1.31–0.29	0.205	−0.19 (0.07)	−1.58–1.55	0.890
Cholesterol	−0.01 (0.00)	−0.02–0.01	0.089	0.03 (0.01)	−0.02–0.03	0.811
NBF1-n-3 PUFA	−2.54 (0.39)	−5.81–5.03	0.042 *	−5.05 (1.98)	−12.83–8.86	0.301
NBF1-n-6 PUFA	−0.43 (0.23)	−0.90–0.04	0.072	−0.45 (0.04)	−1.34–0.43	0.304
NBF1-fiber	0.12 (0.05)	−0.01–0.05	0.853	0.29 (0.23)	−0.41–0.87	0.440
Plant protein	0.78 (0.37)	0.24–1.65	0.592	−0.15 (0.22)	−0.62–0.30	0.485
Animal protein	0.45 (0.12)	0.19–0.71	0.11	0.14 (0.17)	−0.21–0.55	0.423

***** Statistically significant value: *p* = < 0.05. (**A**)**:** CRP, C-reactive protein; PUFA, polyunsaturated fatty acid. (**B**): DBP, diastolic blood pressure; SBP, systolic blood pressure; NBF1, sujiaonori biomaterial-based supplement; n3 PUFA, omega-3 polyunsaturated fatty acids; n6 PUFA, omega-6 polyunsaturated fatty acids.

## Data Availability

Dataset analyzed for this report is can be obtained from the corresponding author (NRN).

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
