# Peer review of "Modulatory Effects of NBF1, an Algal Fiber-Rich Bioformula, on Adiponectin and C-Reactive Protein Levels, and Its Therapeutic Prospects for Metabolic Syndrome and Type-2 Diabetes Patients"

_biomedicines, 2022, doi:10.3390/biomedicines10102572_

Round 1
Reviewer 1 Report
In this study, the authors reported beneficial effects of NBF1, an algal fiber-rich formula, on metabolic syndrome and type 2 diabetes. Clinical cases comprising three subjects with metabolic disorders (two type 2 diabetes mellitus patients, one subject with cardiometabolic risk) are also reported. The clinical observation is valuable; however, the presentation should be substantially improved.
1. Abbreviation (CRP) should not be used in the Title. A suggested title is listed below for the authors’ consideration.
Title: Modulatory Effects of NBF1, an Algal Fiber-rich Bioformula on Adiponectin and C-Reactive Protein Levels, and its Therapeutic Prospects for Metabolic Syndrome and Type-2 Diabetes Patients
2. Both n3-PUFA and adiponectin were elevated in healthy women by NBF1 supplementation, indicating its potential therapy on metabolisms disease. Please add more reasons why the authors chose the NBF1 treatment in the Introduction.
3. Please add the statements of clinical reports (Fig.1) to the Material and Method.
4. Please close the “track changes” for the manuscript.
5. Writing should be substantially improved.
Author Response
COMMENTS FROM REVIEWER1:
In this study, the authors reported beneficial effects of NBF1, an algal fiber-rich formula, on metabolic syndrome and type 2 diabetes. Clinical cases comprising three subjects with metabolic disorders (two type 2 diabetes mellitus patients, one subject with cardiometabolic risk) are also reported. The clinical observation is valuable; however, the presentation should be substantially imporved.
- Abbreviation (CRP) should not be used in the Title. A suggested title is listed below for the authors’ consideration.
Title: Modulatory Effects of NBF1, an Algal Fiber-rich Bioformula on Adiponectin and C-Reactive Protein Levels, and its Therapeutic Prospects for Metabolic Syndrome and Type-2 Diabetes Patients.
-Answer: Thank you. We corrected the title as you suggested.
- Both n3-PUFA and adiponectin were elevated in healthy women by NBF1 supplementation, indicating its potential therapy on metabolisms disease. Please add more reasons why the authors chose the NBF1 treatment in the Introduction.
-Answer: Thank you for this remark. We deleted the sentence related to NBF1 in the introduction section.
- Please add the statements of clinical reports (Fig.1) to the Material and Method.
-Answer: Thanks. It is done.
- Please close the “track changes” for the manuscript.
-Answer: OK; it is done.
- Writing should be substantially improved.
-Answer: OK. We had the revised manuscript edited by English editor.
THANK YOU!
Reviewer 2 Report
This clinical trial investigated the beneficial effects of Sujiaonori algal-biomaterial (NBF1) in healthy adults living in Japan and reported the benefits of NBF1 in 3 patients with metabolic conditions. The study is interesting. However, the manuscript remains many issues requiring further modifications. My comments are as follows: 1. Introduction: The background of this study is not clear. Please briefly introduce about NFB1 and highlight the reasons of your study purposes. 2/ Materials and Methods: This section is not well presented. For example: - Please clarify the inclusion and exclusion criteria. - Please clarify the study design. - Please clarify how to calculate the sample size? What was the sampling method? - Please add the CONSORT workflow. - Page 2, Line 84: 16 in NBF1 group, 15 controls = 31, not 32 female participants. - Page 3, Line 105: Please clarify the origin of Sujiaonori-based supplement. - Page 3, Line 135-136: A paired t-test should be used when each subject has a pair of measurements, such as a before and after score. 3. Results: - Page 4, Line 142: Participants were non-smoking female. How about male and smokers? - Table 1: Did the authors test the distribution of data? Which statistical test was used? 4. Discussion: Since NBF1 was used in humans in this study, please discuss about the bioavailability of this supplement. |
Author Response
COMMENTS FROM REVIEWER2
- Introduction: The background of this study is not clear. Please, briefly introduce about NBF1 and highlights the reasons of your study purpose.
-Answer: Thank you. We clarified the objective of this study in the introduction section. A description of NBF1 is given in the Methods section.
- Materials and methods:
- Clarify the inclusion and exclusion criteria.
- -Answer: Inclusion and exclusion criteria are provided, as well as a diagram that shows the sampling process of the participants following CONSORT guidelines.
- Clarify the study design:
-Answer: we clarified the study design in the revised paper.
2.3. clarify how to calculate the sample size? What was the sampling method?
-Answer: Thank you for mentioning this. We added a diagram showing the sampling process of the participants in this pilot study.
2.4. Please add the CONSORT workflow.
-Answer: Thanks. We added CONSORT flow diagram (Fig. 1).
2.5. Page2, Line84: 16 in NBF1 group, 15 controls=31, not 32 female participants.
-Answer: Thanks for this remark. There was a mistake, and it is corrected: 16 NBF1 and 15 controls.
2.6. Page3, Line 105: please clarify the origin of Sujiaonori-based supplement.
-Answer: We added explanation about the supplement in the Methods section.
2.7. Page3, Line 135-136: A paired t test should be used when each subject has a pair of measurements, such as a before and after score.
-Answer: Thank you for raising this fact. Of course, paired and unpaired t test are used for group comparison when outcome variable is continuous; however, the choice is oriented by the study design, more specifically by the number of comparative groups (2 different groups or 1 group with before/after evaluation data). Additional explanation is provided in the Methods section (data analysis).
- Results:
3.1. Page4, Line 142: Participants were non-smoking female. How about male and smokers?
-Answer: The study site was a university campus; this institution used to be a women university. It has just started to enroll male students recently. In this study, there was 1 male participant enrolled but he was excluded for the analysis. An explanation is given in the Methods section.
3.2. Table 1: did the authors test the distribution of data? Which statistical test was used?
-Answer: Yes, we performed the Shapiro-Wilk test to check for normality of data distribution for outcome variables (selected cardiometabolic markers). Given that data were not normally distributed, linear regression model was used to determine associations with dietary and supplemented (intervention) nutrients. Unpaired and paired t test were used for group comparisons. More explanations are provided in the Methods section.
- Discussion: since NBF1 was used in humans in this study, please discuss about the bioavailability of this supplement.
-Answer: The supplement is food-based; the powder contains 100% of an alga species used in Japanese diet. It is not made of a single chemical compound of the alga for which bioavailability could be evaluated. Nonetheless, the main compound of the biomaterial is ulvan (~65%), a sulfated polysaccharide, known to have high solubility and low bioavailability (Ning et al., Marine Drugs 2022).
THANK YOU!
Round 2
Reviewer 1 Report
The authors claimed to answer all my comments but did not show convincing revision in the revised manuscript. Thus, the manuscript is not recommended for publication.
For example:
- Abbreviation (CRP) should not be used in the Title. A suggested title is listed below for the authors’ consideration.
Title: Modulatory Effects of NBF1, an Algal Fiber-rich Bioformula on Adiponectin and C-Reactive Protein Levels, and its Therapeutic Prospects for Metabolic Syndrome and Type-2 Diabetes Patients.
Answer: Thank you. We corrected the title as you suggested.
However, the title is not changed in the revised manuscript.
- Both n3-PUFA and adiponectin were elevated in healthy women by NBF1 supplementation, indicating its potential therapy on metabolisms disease. Please add more reasons why the authors chose the NBF1 treatment in the Introduction.
-Answer: Thank you for this remark. We deleted the sentence related to NBF1 in the introduction section.
Deletion of the sentence related to NBF1 in Introduction did not provide the reasons why the authors chose the NBF1 treatment.
Author Response
ANSWERS TO REVIEWER1
- Abbreviation (CRP) should not be used in the Title. A suggested title is listed below for the authors’ consideration.
Title: Modulatory Effects of NBF1, an Algal Fiber-rich Bioformula on Adiponectin and C-Reactive Protein Levels, and its Therapeutic Prospects for Metabolic Syndrome and Type-2 Diabetes Patients.
-Answer: we adopted the title suggested by Reviewer1.
- Both n3-PUFA and adiponectin were elevated in healthy women by NBF1 supplementation, indicating its potential therapy on metabolisms disease. Please add more reasons why the authors chose the NBF1 treatment in the Introduction.
-Answer: The corresponding authors have conducted several experimental studies in the past using edible river algal species growing in Japan. Biological changes caused by a number of algal biomaterials could give insights on their bioactive properties and, especially their potential health effects. Given that Japanese “traditional” diet is reported to promote adiponectin production and cardiometabolic health, and considering previous experimental outcomes on metabolic markers such as adipokines, we have targeted algal food. And NBF1 is just biomaterial from on species of japan-grown algae.
Reviewer 2 Report
NO COMMENTS
Author Response
To the reviewer,
Thank you for your review of our paper!
The Authors.
Round 3
Reviewer 1 Report
The revision is acceptable, and thus recommended for publication.